



# Similarities within a multi-model ensemble: functional data analysis framework

Eva Holtanová[1], Thomas Mendlik[2], Jan Koláček[3], Ivanka Horová[3], Jiří Mikšovský[1]

[1]Department of Atmospheric Physics, Faculty of Mathematics and Physics, Charles University, V Holešovičkách 2, Prague, 180 00, Czech Republic
[2]Wegener Center for Climate Studies, University of Graz, Brandhofgasse 5/1, Graz, 8010, Austria
[3]Department of Mathematics and Statistics, Faculty of Science, Masaryk University, Kotlářská 267/2, 611 37, Brno, Czech Republic

*Correspondence to*: Eva Holtanová (Eva.Holtanova@mff.cuni.cz)

**Abstract.** Despite the abundance of available global and regional climate model outputs, their use for evaluation of past and future climate changes is often complicated by substantial differences between individual simulations, and the resulting uncertainties. In this study, we present a methodology framework for the analysis of multi-model ensembles based on functional data analysis approach. A set of two metrics that generalize the concept of similarity based on the behaviour of entire simulated climatic time series, encompassing both past and future periods, is introduced. As far as our knowledge, our method is the first to quantitatively assess similarities between model simulations based on the temporal evolution of simulated values. To evaluate mutual distances of the time series we used two semimetrics based on Euclidean distances between the simulated trajectories and on differences in their first derivatives. Further, we introduce an innovative way of visualizing climate model similarities based on a network spatialization algorithm. Using the layout graphs the data are ordered on a 2-dimensional plane which enables an unambiguous interpretation of the results. The method is demonstrated using two illustrative cases of air temperature over the British Isles and precipitation in central Europe, simulated by an ensemble of EURO-CORDEX regional climate models and their driving global climate models over the 1971–2098 period. In addition to the sample results, interpretational aspects of the applied methodology and its possible extensions are also discussed.

## 1 Introduction

While numerical climate models serve as the cardinal tool of contemporary climatology, their outputs are typically burdened by distinct uncertainties, manifesting through substantial differences between individual simulations. Here, we address the issue of comparing various climate simulations and quantifying their differences by introducing a methodology for analysis of multi-model ensembles and the relationship between nested regional climate model simulation and its driving global climate model run. We propose use of a metric generalizing the concept of similarity, based on the information contained in





the entire simulated climate series, extending from historical to future periods. The evaluation framework is based on functional data analysis (further denoted as FDA; Ramsay and Silverman, 2005, 2007; Ferraty and Vieu, 2006).

The analysis of uncertainties in climate model outputs is a key research topic, especially due to the use of model simulations as inputs for studies of possible future climate changes impacts. The results of the respective analyses serve as the basis for

important adaptation and mitigation decisions, with a critical role belonging to the information on reliability of the projections and the structure of the relevant uncertainties. Climate model outputs are subject to uncertainties coming from various sources, including imperfect initial and boundary conditions, parameterizations of small scale processes or necessary choices and simplifications in the model structure (numerical schemes, spatial resolution, etc.). For detailed discussion see e.g. Tebaldi and Knutti (2007). When considering regional climate models (RCMs), it is necessary to take into account some

additional factors, mainly connected to the limited integration domain (Laprise et al., 2008) or possible inconsistencies of parameterization schemes between driving and nested models (Denis et al., 2002). The estimate of the uncertainties in climate model outputs must accompany any future climate change scenario.

One of the most frequently used ways of uncertainty assessment is the analysis of multi-model ensemble (MME) spread (e.g. Belda et al., 2017; Holtanová et al., 2010; Prein et al., 2011). The main aim of MMEs is to sample the uncertainty stemming

from choices in model structure, parameterization schemes and, in case of RCMs, also boundary conditions. Estimating the uncertainty range based on the MME spread is not a straightforward task, as currently available MMEs suffer from various deficiencies. One obstacle is raised by the deficiencies in the statistical experimental design: Models are developed voluntarily from institutions worldwide. This problem is further amplified when designing an ensemble of RCMs. An RCM is driven by a global climate model (GCM) which has a substantial effect on the nested simulation (Heinrich et al., 2014). It

is not computationally feasible to run all combinations of RCMs with every GCM. Therefore, for a proper uncertainty assessment it is crucial to investigate the interactions between driving GCMs and nested RCMs and their respective influence on the total MME spread (e.g. Holtanová et al., 2014; Heinrich et al., 2014; Holtanová and Mikšovský, 2016).

In addition, climate models (even across developing institutions) are known to share certain components, leading to inter-model similarities and dependencies. This makes it difficult to justify the independence assumption when quantifying the

uncertainty of MMEs with standard statistical models. Recently, innovative methods have been developed to identify groups of similar climate models (e.g. Knutti et al., 2013) and account for the similarities (Annan and Hargreaves, 2017). However, these methods quantify model similarity based on either their behavior in approximating the historical climate or purely on their projected climate change signals. Some studies included evaluation of the relationship between the driving GCM and nested RCM based on more advanced climatic characteristics (e.g. Rajczak and Schär, 2017; Crhová and Holtanová, 2018),

but their approach to the issue was rather qualitative. As far as our knowledge, our method is the first to quantitatively assess similarities between model simulations based on the temporal evolution of simulated values.

To illustrate a possible application of the proposed methodology we analyze (dis)similarities between members of the EURO-CORDEX multi-model ensemble (Jacob et al., 2013) and their driving GCMs. The inter-model distances between the trajectories of the temporal development of running 30-year mean changes in seasonal mean air temperature and



precipitation are evaluated. We first assessed the similarities between ensemble members for time series averaged over eight large European regions defined by Christensen and Christensen (2007) that have been widely used for climate model assessments (e.g. Rajczak and Schär, 2017; Holtanová and Mikšovský, 2016; Mendlik and Gobiet, 2016). Here we show the results for only two chosen cases, namely the winter mean air temperature over the British Isles and mean summer
precipitation over Eastern Europe. These two cases were chosen to illustrate two distinct cases of GCM-RCM interaction.

The paper is structured as follows. In Sect. 2 the EURO-CORDEX regional climate models and their driving global climate models are briefly introduced. In Sect. 3 the methodology is described, including the basic information about the FDA approach. Sect. 4 explains the application of methodology framework and Sect. 5 is devoted to description of the results of the case study. Sect. 6 summarizes key features of the proposed framework and offers possible further applications.

**2 Data**

The methodology framework is presented on the sample of RCM simulations from the EURO-CORDEX initiative (Jacob et al., 2013; http://www.euro-cordex.net/) together with their driving GCMs. We use 13 RCM simulations driven by 9 different GCMs. All RCM simulations have 0.44° horizontal resolution. The RCM simulations were conducted for the period 1951–2100, with some of them starting in 1971 or ending in 2098. We therefore concentrate on the period 1971–2098. After the
year 2006 model simulations incorporated the representative concentration pathway RCP8.5 (van Vuuren et al., 2011). The GCM simulations were performed under the CMIP5 protocol (Taylor et al., 2012). The list of models is given in Table 1 and the GCM-RCM simulation matrix in Table 2. To identify individual simulations, we use the acronyms consisting of RCM and GCM abbreviation (as defined in Table 1) connected with underscore character. In case of driving GCM simulation we use "dGCM" instead of the RCM identification.
We concentrate on running 30-year mean changes in seasonal mean air temperature and precipitation (changes of running 30-year mean averages throughout the period 1971–2098 in comparison to the reference period 1971–2000). For the purpose of introducing the methodology, we only present two illustrative cases: winter mean air temperature changes over the British Isles (denoted as DJF tas over BI, data shown in Fig. 1a) and summer precipitation changes over Eastern Europe (JJA pr over EA, Fig. 2a).

**3 Methodology**

**3.1 Functional data analysis approach**

We analyzed (dis)similarities between the temporal development of simulated 30-year running mean air temperature and precipitation changes. The original dataset consisted of simulated values $y_{ik}$ at central years of the 30-year periods $t_k$, $k = 1,...,K$, ranging from 1986 to 2083 (hence $K = 98$) for each model, $i = 1,...,n$. These sequences of simulations were converted



to functional form using the B-spline basis system $B_j(t)$, $j = 1,...,N$. Each sequence was approximated by a spline function $x_i(t)$ in the form

$$x_i(t) = \sum_{j=1}^{N} c_{ij} B_j(t), i = 1,\ldots,n. \tag{1}$$

The B-splines $B_j(t)$ were polynomials of order four with twenty equally spaced knots, $c_{ij}$ were real coefficients in the B-spline basis. Such use of order four B-splines implied $N = 22$ basis functions. Spline functions $x_i(t)$ were constructed in order to minimize the penalized squared error

$$\sum_{i=1}^{n} \sum_{k=1}^{K} \left[y_{ij} - x_i(t_k)\right]^2 + \lambda \int_{t_1}^{t_K} \left[\frac{d^2}{dt^2} x_i(t)\right]^2 dt \tag{2}$$

with respect to the coefficients $c_{ij}$. The smoothing parameter $\lambda$ was selected via cross-validation method. The cross-validation was based on the minimization of the following expression,

$$\sum_{i=1}^{n} \sum_{k=1}^{K} \left[y_{ij} - x_i(t_k, \lambda, -k)\right]^2, \tag{3}$$

where $x_i(t_k, \lambda, -k)$ denotes the leave-one-out estimator of $x_i(t)$ omitting the $k$-th observation $(t_k, y_{ik})$. The actual calculation is based on minimization of the error of $x_i(t, \lambda, -k)$ using a smoothing operator – see, e.g., Craven and Wahba (1978) for details.

The representative examples of the functional data from panels (a) of Fig. 1 and 2 are depicted in panels (b) of the respective figures.

One of the aims of this study was to explore the first derivative of the response function. Thus, the first derivative curves $x_i'(t)$ were expressed in a similar manner, using the same B-spline basis with coefficients $c'_{ij}$,

$$x_i'(t) = \sum_{j=1}^{N} c'_{ij} B_j(t), i = 1,\ldots,n. \tag{4}$$

All subsequent analyses were conducted separately on both $x_i(t)$ and $x'_i(t)$.

For the representation of functional data in statistical software R (R Core Team, 2013), we used the package fda (Ramsay et al., 2017). It provides several basis options for functional data including B-splines presented above and further functional data processing techniques.

Since the time series analyzed in the present study are relatively smooth, a metric and a semimetric were constructed to represent the distance separation between two curves (note that the smaller the cross-distance, the more similar the two curves are). Such approach seems to be appropriate, see e.g. Pokora et al.(2017). Let $f_1$ and $f_2$ be two curves, specifically two cubic smoothing splines in our case. A well-known and widely-used distance between given curves $f_1$ and $f_2$ is the $L_2$-metric, $d_0(f_1, f_2)$. It is a nonnegative number, whose square is defined as the integral

$$d_0^2(f_1, f_2) = \int_{t_1}^{t_K} [f_1(t) - f_2(t)]^2 dt. \tag{5}$$

Let us call this common metric as $d_0$-distance (Euclidean distance).

Similarly, a common way to build a semimetric between two curves is to consider the $L_2$-distance between the first derivatives of the curves. More precisely, given two curves $f_1$ and $f_2$, we define the $d_1$-distance $d_1(f_1, f_2)$ to be a nonnegative number, whose square is given by the integral





$$d_1^2(f_1, f_2) = \int_{t_1}^{t_K} [f_1'(t) - f_2'(t)]^2 dt. \tag{6}$$

Fig. 3 illustrates examples of two parts of time series that are evaluated as quite different with large distance $d_0 = 112.8$ but similar with relatively small distance $d_1 = 1.56$. The main point is that the values of the semimetrics are inferred solely based on the chosen feature (e.g. Euclidean distance for $d_0$) and are independent of other time series characteristics. In Fig. 3 it is
clearly seen that unlike $d_0$, the $d_1$ semimetric does not take into account the mutual bias of the two time series. It only focuses on the character of their temporal development.

### 3.2 Visualization of the similarities

For visualization of mutual distances based on FDA semimetrics we use layout graphs created using the ForceAtlas2 algorithm (Jacomy et al., 2014) within the Gephi software (https://gephi.org/). In these graphs individual members of the
multi-model ensemble are visualized as nodes (each model simulation corresponding to a single node). The ForceAtlas2 algorithm creates a force directed layout of the underlying data. The network of the nodes is created by simulating a physical system and its movement. The nodes are repulsed from each other in analogy to charged particles. At the same time the edges between the nodes attract them like springs (Jacomy et al., 2014). The iterative procedure of finding the nodes positions results in an equilibrium state which corresponds to the final network.
The interpretation of the layout graphs is straightforward. The closer the nodes are to each other, the lower the mutual distance of corresponding simulations is according to the semimetric of interest. The larger the node the more close neighbours, meaning more similar simulations (with similarity defined by the values of selected semimetric). The edges between nearest 10 % of neighbours are made visible. The colours indicate the driving GCM.

### 4 Application of the methodology

Figs. 1 and 2 illustrate the data used for the presented analysis. The lines are coloured according to the driving GCM and the type of line corresponds to RCM. The purpose of the presented methodology is to describe the structure of the multi-model ensemble based on mutual relationships between simulations over the whole investigated time period and evaluate whether the temporal development of the simulated changes is influenced more strongly by the driving GCM or the nested RCM. The first step is the calculation of mutual distances between the curves corresponding to individual ensemble members using the
FDA semimetrics $d_0$ and $d_1$ defined in Sect. 3. In order to compare two semimetrics with substantially different range, we transform the values to the interval [0,1] in both cases. To facilitate viewing, we display the results in a pixel plot, see Figs. 4 and 5, with a temperature-colour code (or heatmap, with redder colour for larger similarity, brighter colour for smaller similarity).

Figs. 4 and 5 present the values of $d_0$ (panels (a)) and $d_1$ (panels (b)) distances for the two chosen datasets presented in Figs.
1 and 2. Firstly, there are clear differences between the evaluation based on $d_0$ and $d_1$ semimetrics, because each of them is based on different aspects of evaluated curves. It is well apparent from the comparison of maximum distances. In case of JJA



pr over EA (Fig. 5), the $d_0$ distance is the largest for driving HadGEM GCM (dGCM_HadGEM) and ALADIN RCM driven by CNRMCM (ALAD_CNRMCM). These two simulations effectively represent lower and upper bounds of the multi-model ensemble (Fig. 2). On the other hand, according to $d_1$ the most dissimilar time series are GCM simulations by IPSLCM and CNRMCM (Fig. 5b), because their temporal development has largely an opposite sign, even though they do lie "inside" the

multi-model ensemble (Fig. 2).

The second step of the proposed methodology is to quantitatively evaluate and visualize the similarity between simulations and their clustering according to their mutual distances. This would traditionally be done by means of hierarchical cluster analysis which arranges the members of the multi-model ensemble into a dendrogram, as shown for example in Fig. 6 for DJF tas over BI based on $d_1$ (R function heatmap.2 from package gplots was used for the dendrogram creation, see

Supplement). However, the interpretation of the dendrograms might not be straightforward and relatively similar simulations might be assigned to quite remote clusters. In our example (Fig. 6) this is the case for the simulations of HadGEM and CNRM GCMs which are assigned to two remote clusters, even though their mutual $d_1$ distances are among the lowest from the whole ensemble (the same applies to RCM simulations driven by these two GCMs, Fig. 4b). Similar result can be seen in case of CNRM and MIROC5 GCMs. To overcome this hurdle we propose an innovative method of visualization of the

similarities based on evaluated semimetrics distances, the layout graphs (see Sect. 3.2). Figs. 7 and 8 show the layout graphs for the two investigated cases. The main advantage of the layout graphs in comparison to classical dendrograms is that the structure of the ensemble is shown in 2D and therefore the mutual distances are seen easily. The above noted relationships between the HadGEM, MIROC5 and CNRM clusters are easily interpreted using the layout graph (Fig. 7b).

## 5 Case study results

The methodology described in Sect. 3 was applied to the modelled temperature and precipitation changes from the EURO-CORDEX multi-model ensemble and the respective driving GCMs for eight large European domains (Christensen and Christensen, 2007). Here we only show two cases to illustrate the ability of the proposed method to assess the relationships within the members of the multi-model ensemble. These two sample cases, DJF tas over BI and JJA pr over EA, were chosen because they differ in terms of the results obtained by application of the proposed methodology and the results are

quite illustrative.

As we analyze simulations incorporating RCP8.5, which assumes a rise in greenhouse gas concentrations during the whole 21st century, it is not surprising that all models give a rise in DJF near surface air temperature over the BI region throughout this period (Fig. 1). The RCMs tend to give generally lower temperature change than their driving GCMs, except for RCMs driven by CNRMCM, MPIESM and MIROC5. Regarding the simulated changes in summer mean precipitation over the EA

region (Fig. 2), the model simulations disagree on the sign of precipitation change and the multi-model ensemble has quite a large variance. Some RCMs project larger changes than their driving GCMs (e.g. ALADIN driven by CNRMCM), some give smaller changes (RCA4 driven by IPSLCM).



Based on $d_0$, the distances calculated for JJA pr over EA are mostly quite low, lower than 0.25 with a couple of outliers, namely ALAD_CNRMCM and driving simulations of HadGEM and CSIRO (Fig. 5a). The $d_0$ distances for DJF tas over BI are more evenly distributed (Fig. 4a), because there are not so distinct outliers. The $d_1$ distances are higher than $d_0$ values in both regions, and generally higher for JJA pr over EA than for the other case (compare panels (b) in Figs. 4 and 5). That

means that there are less members of the ensemble behaving in a similar manner for the EA case than for the BI case.

Regarding the influence of the driving GCM on the nested RCM simulation, based on both $d_0$ and $d_1$, for DJF tas over BI the simulations driven by the same GCM are more clustered together than in case of JJA pr over EA, which is visible by comparing Figs. 4 and 5 and confirmed in Figs. 7 and 8. The clustering is stronger for $d_1$ results. An evaluation of Fig. 4b reveals that for DJF tas over BI the $d_1$ distance of the RCM simulation and its driving GCM simulation is close to zero in

most cases, as well as the mutual distances of RCA4 simulations driven by the same GCM (e.g. MPIESM, NorESM, CNRMCM). In case of JJA pr over EA (Fig. 5b) the $d_1$ distances tend to be higher and rather independent of the driving GCM. For example, the distance between the simulations of RCA4 and REMO both driven by MPIESM is larger than the distances between RCA4 simulations driven by different GCMs. What we "dig in" for in Figs. 4 and 5 is clearly seen on the first sight in Figs. 7 and 8, respectively. The configuration of the layout graphs confirms a strong clustering according to the

driving GCM in the case of DJF tas over BI and higher degree of interaction between GCM and RCM in case of JJA pr over EA (compare the corresponding panels in Figs. 7 and 8).

It is clearly seen that when large-scale phenomena are responsible for output, as in case of temperature changes over BI region, RCMs tend to be very close to driving GCM, and different GCMs are apart from each other (Figs. 1 and 7). On the contrary, when smaller scale processes are more in play, such as in case of JJA precipitation changes over EA, the results are

more influenced by RCMs (Figs. 2 and 8). This does not automatically imply any real added value in the sense of more realistic simulation. Rather, it points to differences in implementation of the local processes in different RCMs. In our case, different parameterization schemes employed to simulate convection, microphysical processes in clouds and surface processes including soil moisture are possible candidates.

Regarding the three RCM simulations driven by CNRMCM GCM (RCMs denoted here as ALAD, CUNI and RCA4), it has

been recently revealed that the boundary conditions for the historical period have been flawed with an inconsistency (personal communication with members of the EURO-CORDEX community). Specifically, 2D and 3D fields provided to the RCMs come from different members of the ensemble of CNRMCM simulations with perturbed initial conditions and therefore they are mutually out of phase. However, our results do not show any anomalous behaviour of these simulations. When we calculated the distances for the curves for first twenty 30-year periods (i.e., those with the central year before 2005,

which is the end of the historical period) and for the last 20-year periods, we found out that the distance of RCM simulations driven by CNRMCM and their driving GCM is smaller for the future period than for the reference one (not shown). That is probably partly caused by above mentioned discrepancies in the boundary conditions, but the effect is rather small.



## 6 Discussion and conclusions

We have presented an innovative methodology for assessment of the structure of the multi-model ensemble and mutual relationships between its members. A case study evaluating the similarities within the EURO-CORDEX multi-model ensemble extended by the driving CMIP5 GCM simulations has been performed. Attention has been paid especially to the

relationship between the driving GCM and nested RCM simulations in terms of temporal development of simulated temperature and precipitation changes over two European regions. Contrary to previous studies, the assessment takes into account not only simulated values for a certain time period (reference or future), but the character of the simulated temporal development of studied variables as a whole. This is done by generalization to functional similarity of the time series. To evaluate mutual distances of the time series we used two semimetrics based on the Euclidean distances between the

simulated trajectories ($d_0$) and on differences in their first derivatives ($d_1$). The similarity between an RCM and its driving GCM points to a strong forcing and rather low influence of RCM on the simulations of temporal development of the variable of interest. The $d_1$ distances are bias invariant while similarity evaluated by $d_0$ is largely influenced by common biases of model simulations. A small $d_1$ mutual distance between two simulations does not automatically imply similarity in climate change signal for a selected time period, it rather means that the shape of the temporal development is similar.

In general, the $d_0$ similarity indicates agreement in bias and climate change signal, which is influenced by various feedbacks in the climate system and which might be differently pronounced in different models. The $d_1$ similarity points to similar rate (speed and sign) of climate change in time which is partly modulated by internal variability of the models which again is governed by feedbacks and nonlinearities in climate system.

Furthermore, we presented a new way to visualize climate model similarities, based on a network spatialization algorithm.

Instead of arranging the data in a one-dimensional incremental way (like in case of hierarchical cluster analysis resulting in dendrograms), the data are ordered on a 2-dimensional plane using the layout graphs, which enables an unambiguous interpretation of the results. The interpretation is only made harder by the fact that the graph can be rotated subjectively, the algorithm (see Sect. 3.2) only places each data node relatively to all other nodes, but no absolute coordinate system is defined. Even so, it is a very illustrative way of visualization of the mutual distances between the members of a multi-model

ensemble.

The methodology could be extended to include more climatic variables. Similarly, time series with different temporal aggregation (e.g. monthly or annual time series) could be used as input for the analysis. The results of multivariate evaluation of the similarities and relationships within the multi-model ensemble could be a basis for selection of representative models to be used in impact studies. Previously proposed procedures, such as in Mendlik and Gobiet (2016) or

Herger et al. (2018), could be modified to use the FDA similarities introduced here.

Presented methodology does not take model performance explicitly into account. However, the influence of model quality on similarity is implicitly included. Worse performing models will likely be further away from good models. Furthermore, common modelling deficiencies can lead to common similarities in the validation statistics, and the metric used can account





for it. A dissimilarity between the driving GCM and the nested RCM simulations can point to a situation where the GCM does not simulate a certain physical process correctly while the RCM improves it. Moreover, the methodology can be easily modified to serve as a mean of model performance evaluation through performing the analysis for the reference period and including the observed time series. In that case, the results could be used for definition of model weights and calculation of

weighted multi-model mean. For example, in Sanderson et al. (2017) the model weights are based on inter-model distance matrices with the distances defined by root mean square difference (RMSD) between the simulations. The FDA similarities between model simulations could be used instead of the RMSD. Similarly, the inter-model distances, if calculated for the whole CMIP5 GCM ensemble, could serve as a basis for the analysis of inter-model dependencies, as recently discussed for example in Annan and Hargreaves (2017). Finally, it can be mentioned that the presented methodology could be extended by

using the functional principle component analysis (PCA). Nowadays, the functional PCA is a very popular and powerful exploratory technique. Its applications on real data indicate that it could further improve our results.

*Code and/or data availability.* The analysis have been conducted within the R environment and using the Gephi software, which are both freely available. The R code is made available in the Supplement of this paper (contained in the Rcode.R

together with npfda.R from Ferraty and Vieu (2006), available at https://www.math.univ-toulouse.fr/~ferraty/SOFTWARES/NPFDA/index.html). The underlying data are available via ESGF infrastructure (https://www.earthsystemcog.org/projects/cog/). The time series of running 30-year mean temperature and precipitation changes used in the presented case study are available in the form of .RData files in the Supplement to this paper. The input files for Gephi software can be prepared using the Rcode.R and prepare_graphs.py.

*Competing interests.* The authors declare that they have no conflict of interest.

*Acknowledgements.* This work has been supported by project UNCE 204020/2012 (Charles University), project MUNI/A/1204/2017 (Masaryk University), STARC-Impact B567181 (ACRP 8th call, Austrian Climate and Energy

Funds KLIEN) (University of Graz) and Mobility project 8J18AT017.

We acknowledge the World Climate Research Program's Working Group on Regional Climate and the Working Group on Coupled Modeling, the coordinating bodies behind CORDEX and CMIP5. We also thank the climate modeling groups (listed in Table 1 of this paper) for producing and making available their model output. We also acknowledge the Earth System Grid Federation infrastructure, an international effort led by the U.S. Department of Energy's Program for Climate

Model Diagnosis and Intercomparison, the European Network for Earth System Modeling, and other partners in the Global Organisation for Earth System Science Portals (GO-ESSP).




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



| Acronym | Type | Model ID | Institute |
|---------|------|----------|-----------|
| CCLM | RCM | CCLM4-8-17 | Climate Limited-area Modelling Community (CLM-Community) |
| RCA4 | RCM | RCA4 | Swedish Meteorological and Hydrological Institute, Rossby Centre |
| ALAD | RCM | ALADIN53 | Centre National de Recherches Meteorologiques |
| CUNI | RCM | RegCM4 | Charles University |
| CanESM | GCM | CanESM2 | Canadian Centre for Climate Modelling and Analysis |
| CNRMCM | GCM | CNRM-CM5 | Centre National de Recherches Meteorologiques, Meteo-France; Centre Europeen de Recherches et de Formation Avancee en Calcul Scientifique |
| CSIROx | GCM | CSIRO-Mk3.6.0 | CSIRO; Queensland Climate Change Centre of Excellence |
| GFDLES | GCM | GFDL-ESM2M | NOAA Geophysical Fluid Dynamics Laboratory |
| HadGEM | GCM | HadGEM2-ES | Met Office Hadley Centre |
| IPSLCM | GCM | IPSL-CM5A-MR | Institut Pierre Simon Laplace |
| MIROC5 | GCM | MIROC5 | University of Tokyo; National Institute for Environmental Studies Agency for Marine-Earth Science and Technology |
| MPIESM | GCM | MPI-ESM-LR | Max Planck Institute for Meteorology |
| NorESM | GCM | NorESM1-ME | Norwegian Climate Centre |

**Table 1.** List of regional climate models and driving global climate models incorporated in the present study. The first column contains the
5 acronyms used throughout the text. Type column indicates whether the model is regional (RCM) or global (GCM).


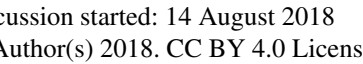


| | | Driving global climate models | | | | | | | | |
|---|---|---|---|---|---|---|---|---|---|---|
| | | CanESM | CNRMCM | CSIROx | GFDLES | HadGEM | IPSLCM | MIROC5 | MPIESM | NorESM |
| **Regional climate models** | **CCLM** | | | | | | | | x | |
| | **RCA4** | x | x | x | x | x | x | x | x | x |
| | **ALAD** | | x | | | | | | | |
| | **CUNI** | | x | | | | | | | |
| | **REMO** | | | | | | | | x | |

**Table 2.** Matrix of regional climate model simulations and their driving global climate models incorporated in the present study.





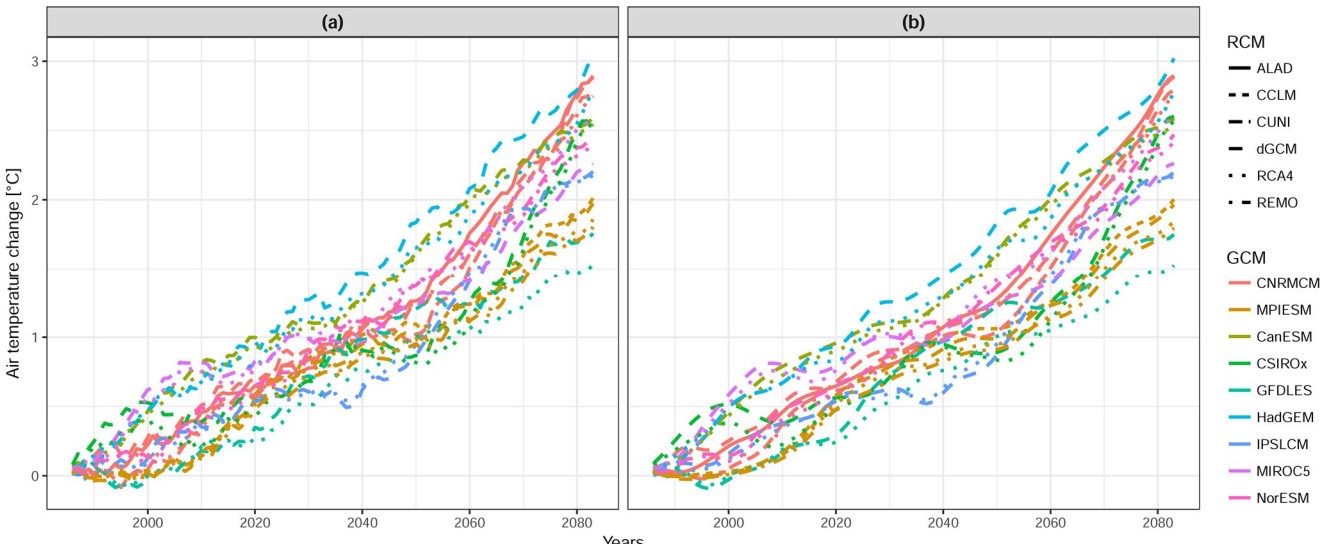

**Figure 1.** (a) Temporal development of running 30-year mean changes in winter (DJF) mean air temperature (changes of running 30-year mean averages throughout the period 1971–2098 in comparison to the reference period 1971–2000) averaged over the British Isles region. (b) Smoothed functional data from panel (a), created as described in Sect. 3. The lines in both panels are coloured according to the driving global climate model (GCM) and the type of line corresponds to regional climate model (RCM). The acronyms of the model simulations are explained in Sect. 2, "dGCM" stands for the driving global climate model simulation.





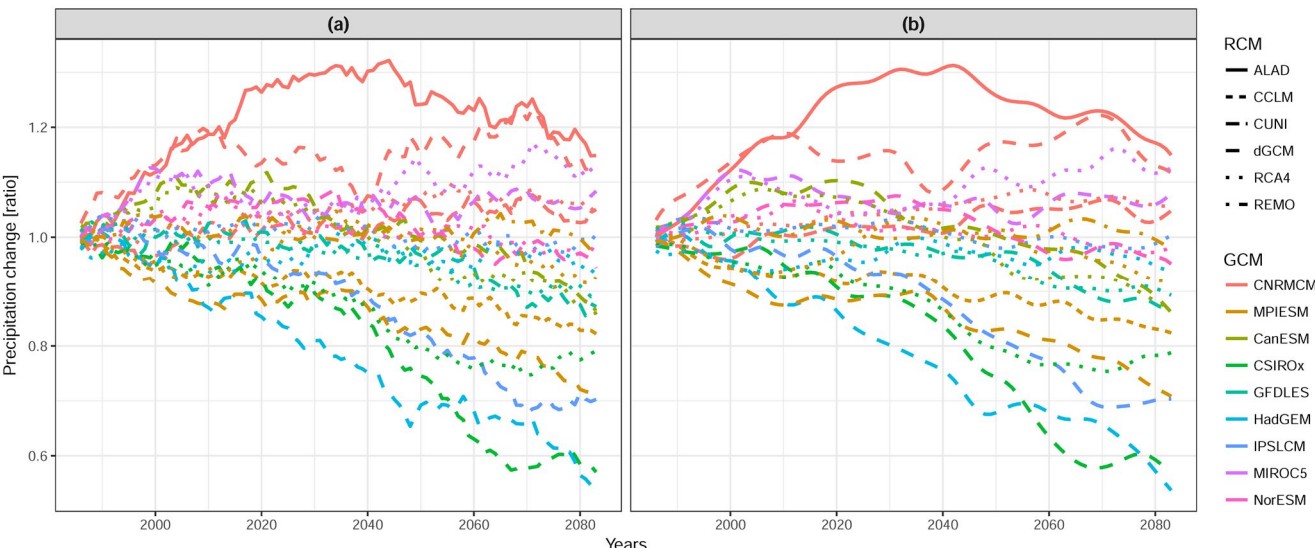

**Figure 2.** The same as Fig. 1, but for running 30-year mean changes in summer (JJA) mean precipitation (relative changes of running 30-year mean averages throughout the period 1971–2098 in comparison to the reference period 1971–2000) averaged over the Eastern Europe region.





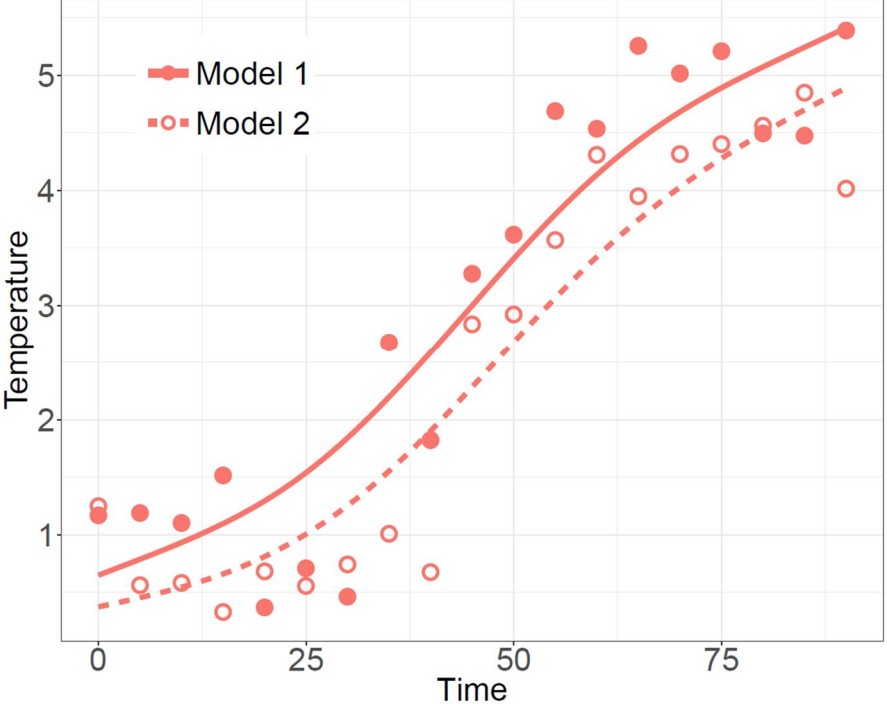

**Figure 3.** Illustration of the functional data analysis approach to evaluation of time series similarity. The two arbitrarily chosen time series shown here (Model 1 and 2) are evaluated as quite different based on $d_0$ but similar based on $d_1$.





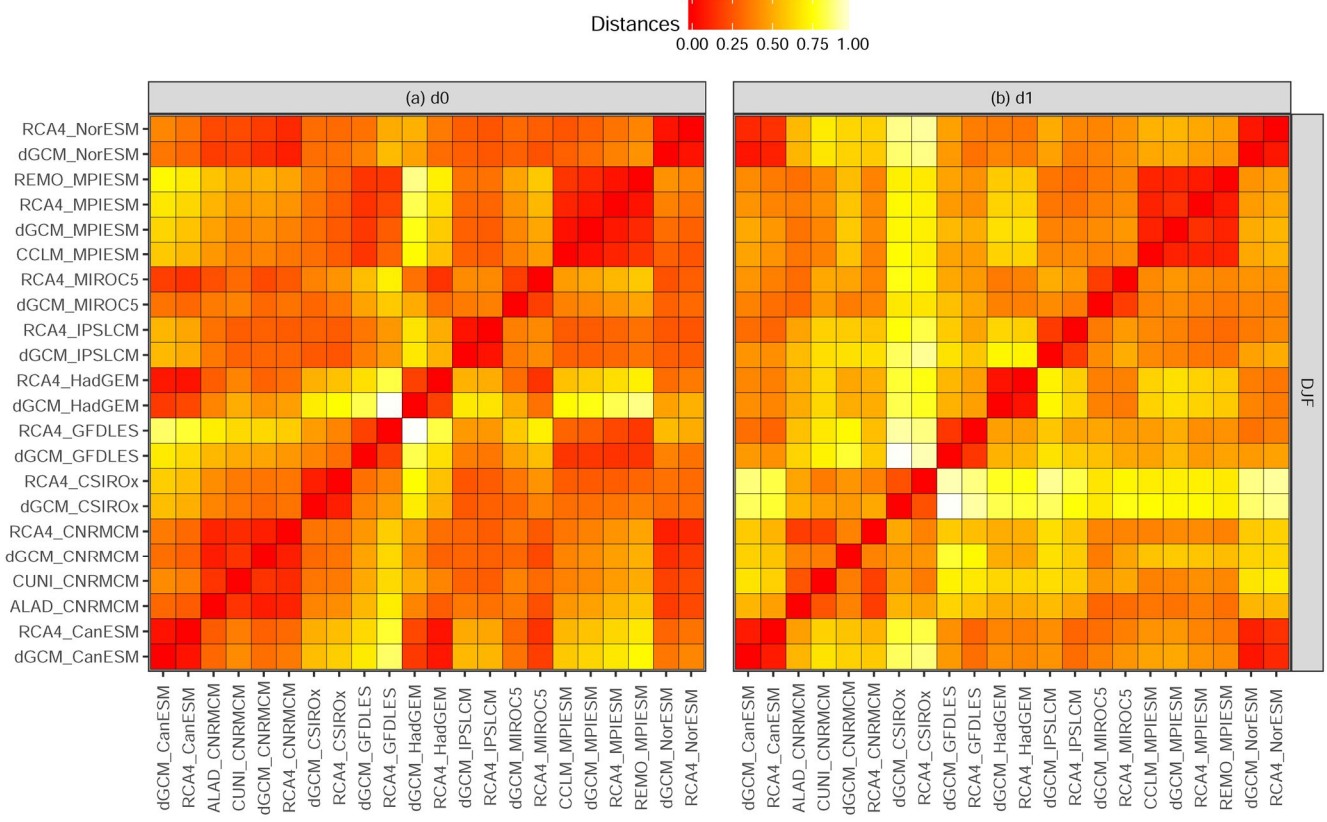

**Figure 4**. (a) Heatmap of the $d_0$ distances for running 30-year mean changes in winter (DJF) mean air temperature over British Isles (the curves shown in Fig. 1b, underlying data in Fig. 1a) with redder colour for larger similarity, brighter colour for smaller similarity between respective curves. The values of the semimetric $d_0$ are scaled to the interval [0,1]. The acronyms of the model simulations are explained in Sect. 2. The definition of the distances is explained in Sect. 3.1. (b) The same as (a), but for $d_1$ distances.

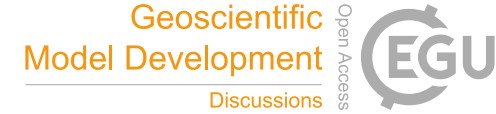



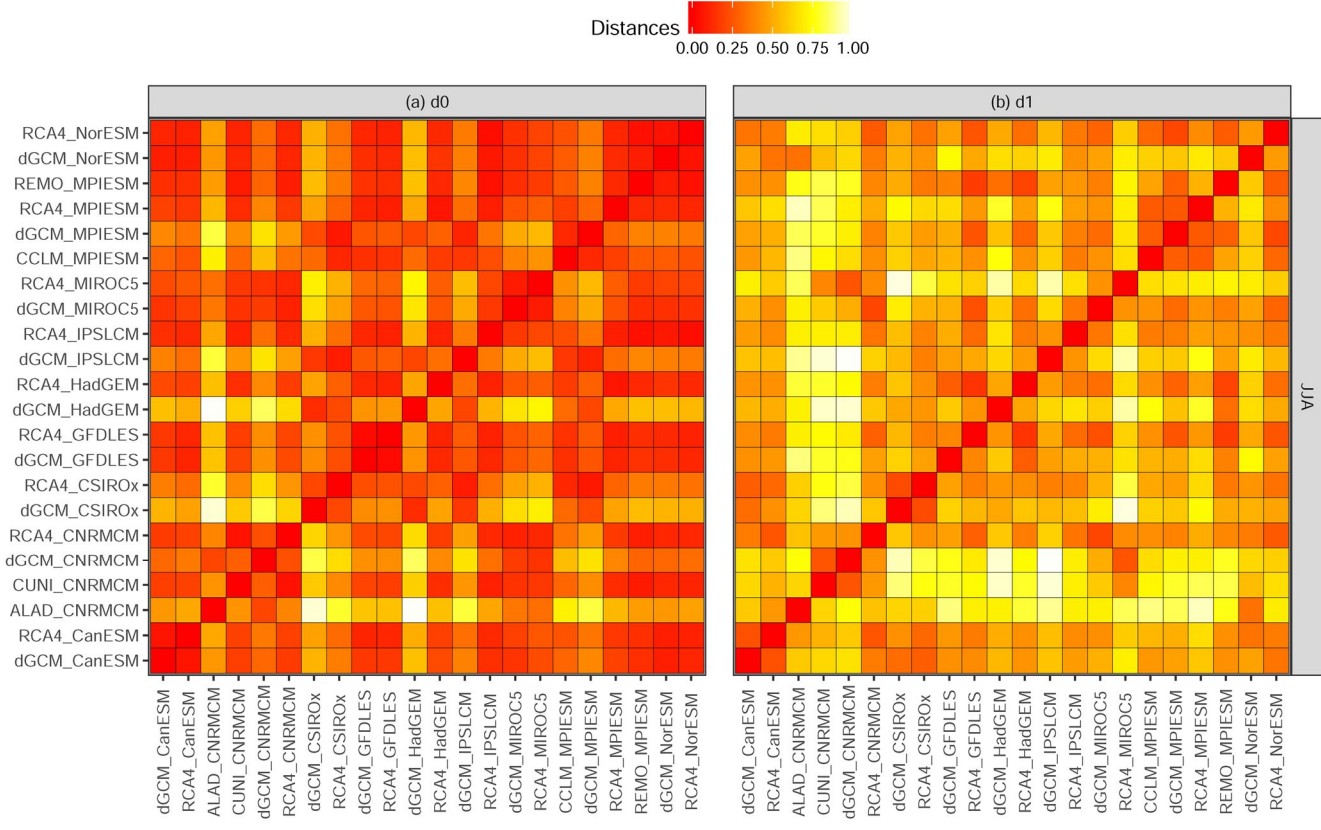

**Figure 5**. The same as Fig. 4, but for running 30-year mean relative changes in summer (JJA) mean precipitation over Eastern Europe region (the curves shown in Fig. 2b, underlying data in Fig. 2a).





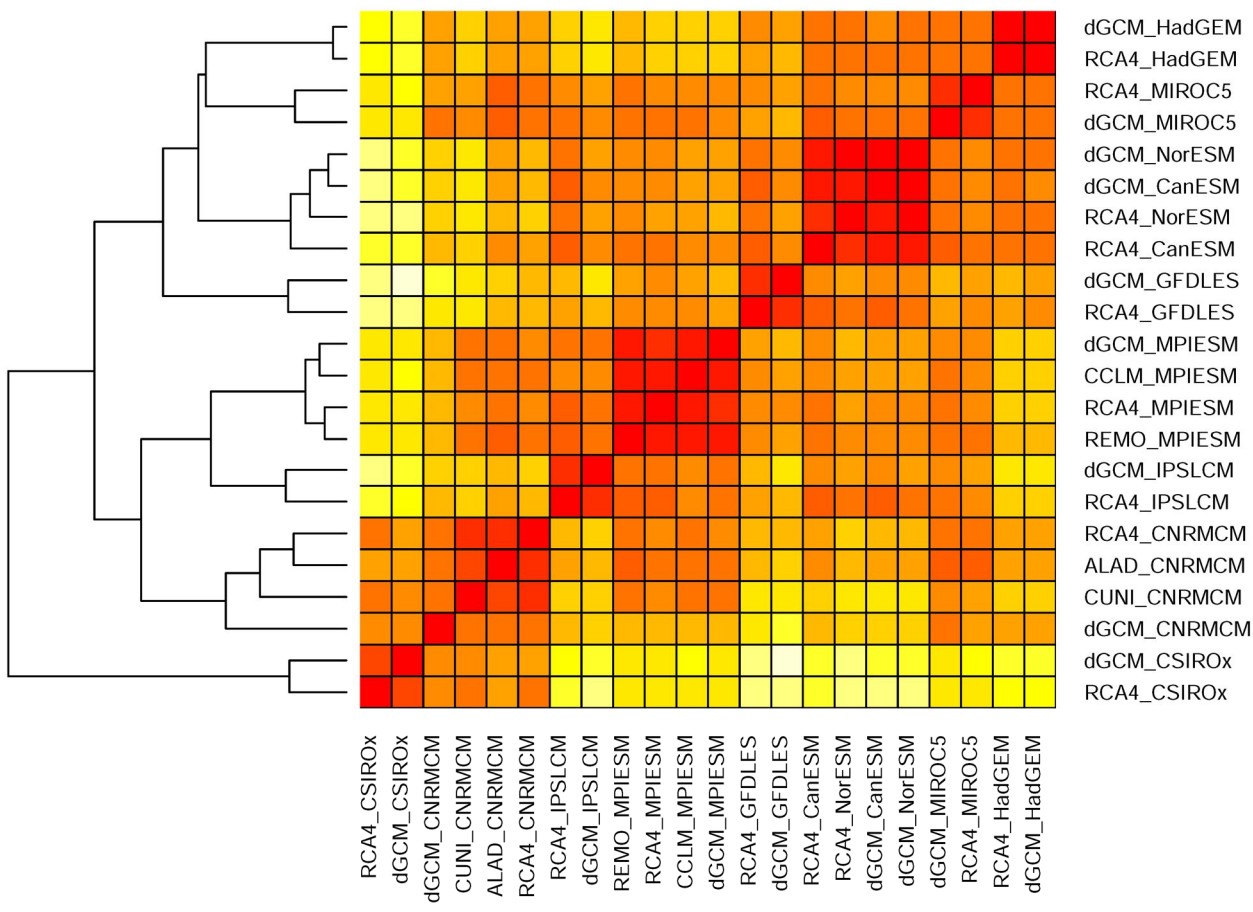

**Figure 6.** An example of the dendrogram resulting from hierarchical cluster analysis based on $d_1$ distances for running 30-year mean changes in winter (DJF) mean air temperature over British Isles (underlying similarity matrix in Fig. 4b).





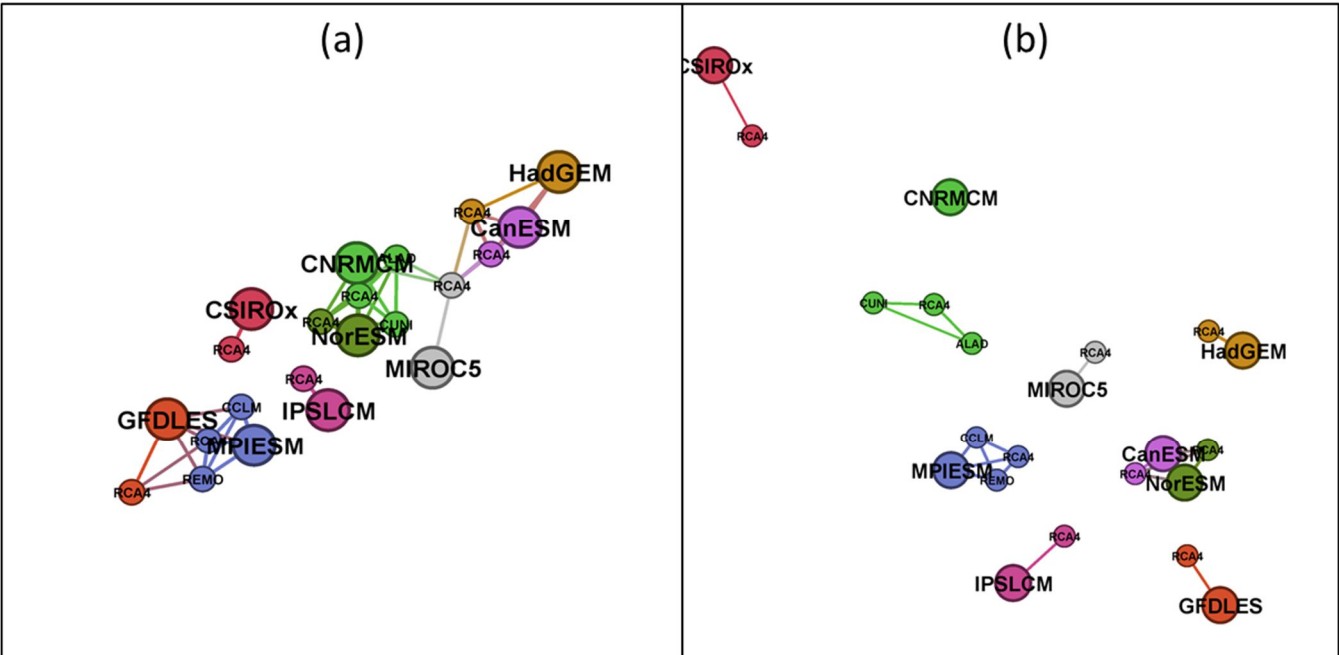

**Figure 7.** (a) Layout graph based on $d_0$ distances for running 30-year mean changes in winter (DJF) mean air temperature over the British Isles (underlying similarity matrix in Fig. 4a). (b) The same as (a), but for $d_1$ distances (underlying similarity matrix in Fig. 4b).



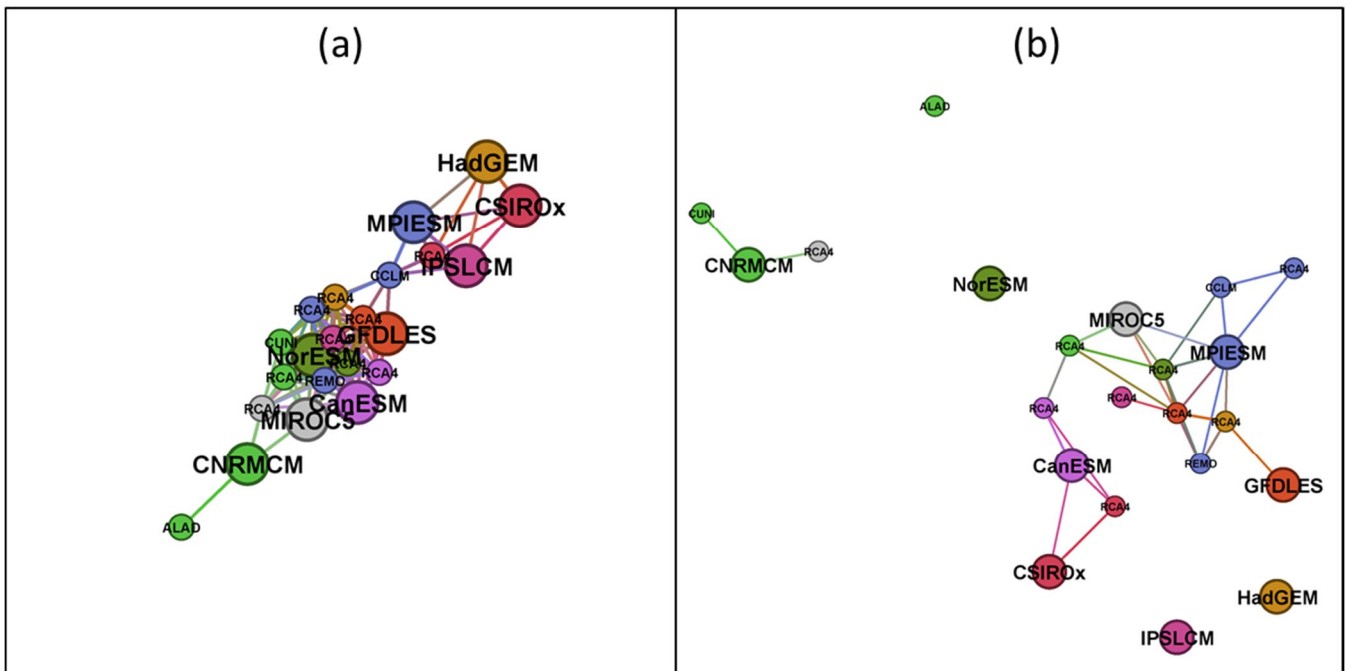

**Figure 8.** The same as Fig. 7, but for running 30-year mean relative changes in summer (JJA) mean precipitation over Eastern Europe region (underlying similarity matrices in respective panels of Fig. 5).