# Peer review of "Similarities within a multi-model ensemble: functional data analysis framework"

_Geoscientific Model Development, 2018_

## Referee Comment (RC1) · Anonymous Referee #1 · 17 Sep 2018

General comments: In the paper "Similarities within a multi-model ensemble: functional data analysis framework" a new method to analyse multi-model ensembles is presented. The analysis is based on the EURO-CORDEX ensemble and focuses the relationship within the ensemble and their driving global climate models. The paper touches a very interesting and relevant topic. However, from my point of view it seems unfinished. The analysis is quite shallow and it is unclear how the new method is adding value to pre-existing studies. Also references to important earlier studies (e.g. Deque et al., 2012) are missing and the relation of the present study to these works is missing. I think it would make the paper much more relevant if the proposed comparison/evaluation with observations (see chapter 6) would be included into the current study. Therefore, I suggest "major revisions" for this paper.

Specific comments: Chapter 3.1: How much do the results depend on the chosen smoothing method and especially on the functional smoothing (e.g., instead of the smoothing one could also use the 30 year running mean - which is already smoothed - and temporal correlation)? Chapter 5: The results of the case studies are very similar to earlier findings (e.g. Deque et al., 2012). Where does the proposed method add value to the earlier findings?

Technical corrections: REMO is missing in Table 1. In Figures 4, 5 and 6 I would suggest to leave out the diagonal (bottom left to top right) and results above or underneath the diagonal because the information is trivial/redundant.

References: Deque et al. (2012): The spread amongst ENSEMBLES regional scenarios: regional climate models, driving general circulation models and interannual variability Climate Dynamics, 2012, 38, 951-964

---

## Referee Comment (RC2) · Anonymous Referee #2 · 2 Oct 2018

The submitted paper describes an inter-model distance metric which uses transient climate evolution to assess model similarity - which could potentially be developed as the basis model weighting scheme which had the capacity to assimilate inter-model distance information. The paper has two main novelties over existing work - a consideration of the distances between GCM/RCM pairs in the CORDEX ensemble and the fact that transient rather than mean climate information is used as the basis for the method. The method is applied to single timeseries of area averaged temperature and precipitation in the CORDEX ensemble The distances are evaluated in a basis set comprising a set of spline-based functions, and take into account both absolute difference in time-series and gradient similarity.

The paper makes advances on the literature in attempting to apply a similarity criterion

to GCM/RCM pairs in CORDEX, and initial results are promising - but need a little more expansion. Results for temperature give qualitatively very different structure to the precipitation results, which needs more discussion. The layout graphs are nice - but the authors should reference relevant earlier work. The paper would be acceptable with minor corrections.

Specific points: - the analysis uses only spatially averaged time-series information, unlike earlier work (e.g. Knutti 2013, Sanderson 2015) which primarily use spatial bias correlation to assess similarity. By not using spatial information, it seems like the authors are throwing away a lot of potentially useful information. This is not a show-stopper - but the authors should acknowledge that by using both spatial and temporal information, more meaningful results could probably be obtained - some parameter sensitivity is required - or at least an explanation of why some arbitrary decisions were made. The domain averaging size, for example - a larger averaging area for precipitation might result in a less noisy field in which model similarities are more accurately identified. Similarly, the averaging period and the parameters of the spline expansion - how sensitive is the method to these choices? - what is the expected noise from climate variability, and can this be quantified more accurately? Can the authors use initial condition ensemble members to identify the expected intermodel distance which arises from climate variability alone? - the graph plots are nice - but there are precedents in the literature for presenting model similarities in 2D space, which should probably be cited here (Sanderson 2015). - I feel slightly more could be made of the discussion of parent GCMs and embedded RCMs. Figure 4 suggests that the parent GCMs dominate the inter-model distances for both d0 and d1 for temperature, but perhaps not for precipitation where there is clear structure from RCM pairs. This is perhaps one of the more interesting results from the paper - and the authors should make more of it. Why is this the case, what are the mechanisms? What recommendations would the authors give for end-users of CORDEX given this finding?

Sanderson, B. M., Knutti, R., & Caldwell, P. (2015). Addressing interdependency in a

multimodel ensemble by interpolation of model properties. Journal of Climate, 28(13), 5150-5170. Knutti, R., Masson, D. and Gettelman, A., 2013. Climate model genealogy: Generation CMIP5 and how we got there. Geophysical Research Letters, 40(6), pp.1194-1199.

———————————————

---

## Author Comment (AC1) · 23 Oct 2018

We are grateful for the evaluation of our paper and all the useful comments and suggestions. We have considered all of them and revised the manuscript accordingly. Please find enclosed the responses, the revised manuscript is submitted as a supplement to a separate author comment.

Please also note the supplement to this comment:
https://www.geosci-model-dev-discuss.net/gmd-2018-157/gmd-2018-157-AC1-supplement.pdf
* * *
[Figure]

2018.

[Figure]

**Supplement:**

We are grateful for the evaluation of our paper and all the useful comments and suggestions. We have considered all of them and revised the manuscript accordingly. Please find below our responses, the revised manuscript is submitted as a supplement to a separate author comment.

The analysis is quite shallow and it is unclear how the new method is adding value to pre-existing studies. Also references to important earlier studies (e.g. Deque et al., 2012) are missing and the relation of the present study to these works is missing. I think it would make the paper much more relevant if the proposed comparison/evaluation with observations (see chapter 6) would be included into the current study.

The comparison with the observations is out of the scope of our study. We concentrate on simulated time series 130 years long, the observations cover only 45 years of it. We chose to show results only for two European regions, as they were interesting and illustrative. But for model skill it would probably be interesting to show different regions, and the study would get disaggregated. Moreover, we pay attention mainly to the structure of the multi-model ensemble and overall uncertainty, independently of model skill, even though, as mentioned in the paper, it can be expected that the better the models, the closer to each other.
Therefore we have not added the comparison with observations to present study, mainly because we think that the study would become disaggregated and would lose clarity.
We have added citations of Déqué et al. (2007, 2012) in the Introduction and to the last section of the paper.

Chapter 3.1: How much do the results depend on the chosen smoothing method and especially on the functional smoothing (e.g., instead of the smoothing one could also use the 30 year running mean - which is already smoothed - and temporal correlation)?

The results do not strongly depend on the smoothing. The dependence is slightly stronger for d1, but even for that the structure of the distances is quite stable for the whole ensemble. We added a comment on this to the end of the Sections 3.1: *"The mutual distances of the curves do not strongly depend on the smoothing parameter, as shown in Fig. S2.1 and S2.2 (see Supplement 2)."* The Fig. S2.1 and S2.2 show the results for an arbitrarily chosen example.

Chapter 5: The results of the case studies are very similar to earlier findings (e.g. Deque et al., 2012). Where does the proposed method add value to the earlier findings?

The aim of our study is not to really reveal new findings regarding the uncertainty of RCM outputs. Rather, we show a new methodology framework and illustrate its usage on a case study. The advantages of the new methodology based on modern statistical approach are discussed in the paper. Regarding the comparison to Déqué et al. (2012), we added a paragraph to the last section: *„Previously, in PRUDENCE and ENSEMBLES projects (predecessors of Euro-CORDEX), the studies of uncertainty and GCM-RCM interactions (mainly Déqué et al., 2007 and Déqué et al, 2012) relied on the analysis of variance of the multi-model ensemble. Quite straightforward and clearly interpretable results suffered from additional uncertainty connected to the necessity to fill in values for missing RCM-GCM pairs with some statistical approach. The methodology proposed in present paper overcomes this issue and uses only the outputs of dynamical models that are available. Further, as already mentioned above, the FDA similarities evaluate the whole simulated time series and are not limited to a reference or future time period. "*

Technical corrections: REMO is missing in Table 1.

We have corrected the Table 1.

In Figures 4, 5 and 6 I would suggest to leave out the diagonal (bottom left to top right) and results above or underneath the diagonal because the information is trivial/redundant.

We have changed the Fig. 4 and 5 as suggested. Fig. 6 includes the dendrogram structure, and the R function used for its creation does not allow leaving out the redundant part.  Therefore we could not change the Fig. 6.

---

## Author Comment (AC2) · 23 Oct 2018

We are grateful for the evaluation of our paper and all the useful comments and suggestions. We have considered all of them and revised the manuscript accordingly. Please find enclosed our responses to specific comments, the revised manuscript is submitted as a supplement to a separate author comment.

Please also note the supplement to this comment:
https://www.geosci-model-dev-discuss.net/gmd-2018-157/gmd-2018-157-AC2-supplement.pdf

**Supplement:**

We are grateful for the evaluation of our paper and all the useful comments and suggestions. We have considered all of them and revised the manuscript accordingly. Please find below our responses to specific comments, the revised manuscript is submitted as a supplement to a separate author comment.

The analysis uses only spatially averaged time-series information, unlike earlier work (e.g. Knutti 2013, Sanderson 2015) which primarily use spatial bias correlation to assess similarity. By not using spatial information, it seems like the authors are throwing away a lot of potentially useful information. This is not a showstopper - but the authors should acknowledge that by using both spatial and temporal information, more meaningful results could probably be obtained

It is certainly true that the evaluation of spatial simulated fields is important. But in current study we have chosen to concentrate on temporal behaviour of the time series averaged over the large European regions. Comparison of spatial fields from RCMs and GCMs is complicated, mainly by large differences in spatial resolution and also by differences in effective spatial resolution (which depends on numerical methods incorporated in the models). We have not figured out how the spatial information could be incorporated in our current setting of the methodology. Spatial fields from GCMs are much smoother than RCMs, and therefore if we convert the fields into functions, the results will be very different in nature. By smoothing (regridding) the RCM fields to GCM-like coarse resolution would result in throwing away a lot of information. But it is probably a good topic for another possible application of our methodology framework, to apply it for evaluation of spatial simulated fields, but for an ensemble consisting of simulations with comparable spatial resolution.

some parameter sensitivity is required - or at least an explanation of why some arbitrary decisions were made. The domain averaging size, for example - a larger averaging area for precipitation might result in a less noisy field in which model similarities are more accurately identified. Similarly, the averaging period and the parameters of the spline expansion - how sensitive is the method to these choices?

The domains used in our study are the quite large "PRUDENCE" regions very often used for analysis of RCM outputs over Europe. The results for smaller regions would probably be more influenced by internal variability and differences between RCMs connected to smaller scale processes and orography representation.
Regarding the averaging period, we have not evaluated the sensitivity of the method. The choice of the length of the period is not basically an arbitrary choice, but it originates in the fact, that we intended to work with long-term means as the main characteristics of climate. And 30-year period is as far as our knowledge the most common period length used in climatology.
Regarding the parameters of the spline expansion, we have analysed the sensitivity of $d_0$ and $d_1$ distances on the amount of smoothing of the underlying curves. The results for an arbitrarily chosen example are shown in Supplement2 and commented on in the end of Section 3.1. The results do not strongly depend on the smoothing. The dependence is slightly stronger for $d_1$, but even for that the structure of the distances is quite stable for the whole ensemble.

what is the expected noise from climate variability, and can this be quantified more accurately? Can the authors use initial condition ensemble members to identify the expected intermodel distance which arises from climate variability alone?

The influence of internal variability on RCM simulation is difficult to be evaluated, as simulations with perturbed initial conditions are not available (as far as our knowledge). Earlier findings (Déqué et al., 2007, Déqué et al., 2012, Hawkins and Sutton, 2009, 2010) suggest that the influence of internal variablity on the overall uncertainty of simulated air temperature and precipitation changes is expected to be rather low. To investigate the issue we compared the results for the ensemble used in

our study with a mini-ensemble consisting of 5 simulations of CNRM-CM5 GCM with perturbed intial conditions (runs denoted as r1i1p1, r10i1p1, r2i1p1,r4i1p1, r6i1p1). We chose this GCM to maximize the number of RCMs driven by it and the extent of resulting mini-ensemble. The figures are available in Supplement3 and the results are commented on in the last section of the revised paper:

*„As explained in the Introduction, the spread of multi-model ensembles is considered as an estimate of structural model uncertainty. For analysis of the influence of internal variability on the overall uncertainty, simulations with perturbed initial conditions can be used. Unlike GCMs, for RCMs these are not generally available. In Supplement3 a suite of figures showing FDA similarities between 5 simulations of CNRM GCM with perturbed initial conditions is provided. The aim of these figures is to illustrate the range of uncertainty stemming from internal variability. We chose CNRM GCM to maximize the number of RCMs driven by this GCM and the number of mini-ensemble members. The figures suggest that for air temperature changes the spread of the CNRM mini-ensemble covers almost a half of the multi-model ensemble spread (Fig. S3.1). In case of precipitation, the portion of the spread is smaller (Fig. S3.2). The $d_0$ and $d_1$ distances between the members of CNRM mini-ensemble are shown in Fig. S3.3 – S3.6. To enable the comparison with the distances for the multi-model ensemble, their values before normalization are provided in Fig. S3.7-S3.10. For air temperature, the maximum inter-model distances are almost twice as large as the inter-simulation distances within the CNRM mini-ensemble (compare Fig. S3.3, S3.4 and S3.7, S3.8). In case of precipitation, the $d_0$ distances between the simulations with perturbed initial conditions are very small in comparison to inter-model distances (Fig. S3.5 and S3.9). However, for $d_1$ distances the difference is not so struggling (Fig. S3.6 and S3.10). The fact that the range of uncertainty connected to internal variability is relatively larger (in comparison to structural uncertainty) for air temperature than for precipitation probably points to larger overall structural uncertainty in case of precipitation than air temperature, i.e. the inter-model differences in simulation of processes connected to precipitation changes are larger than in case of air temperature changes. However, we have to keep in mind that presented results rely only on a limited number of simulations from one GCM."*

Déqué, M., Rowell, D.P., Lüthi, D., Giorgi, F., Christensen, J.H. et al., 2007. An intercomparison of regional climate simulations for Europe: assessing uncertainties in model projections. Climatic Change, 81, Supplement 1, 31–52.

Déqué, M., Somot, S., Sanchez-Gomez, E., Goodess, C. M., Jacob, D., Lenderink, G., Christensen, O. B. (2012): The spread amongst ENSEMBLES regional scenarios: regional climate models, driving general circulation models and interannual variability Climate Dynamics, 2012, 38, 951-964

Hawkins, E., Sutton, R., 2009. The potential to narrow uncertainty in regional climate predictions. Bulletin of the American Meteorological Society. DOI: 10.1175/2009BAMS2607.1

Hawkins, E., Sutton, R., 2010: The potential to narrow uncertainty in projections of regional precipitation change. Climate dynamics. DOI: 10.1007/s00382-010-0810-6

the graph plots are nice - but there are precedents in the literature for presenting model similarities in 2D space, which should probably be cited here (Sanderson 2015).

We have added a comment to the last section:

*"Unlike similar approach of multidimensional scaling used in Sanderson et al. (2015), which also results in 2-dimensional visualization of inter-model distances, the layout graphs do not require defining any data node as a central (reference) point of the whole ensemble."*

I feel slightly more could be made of the discussion of parent GCMs and embedded RCMs. Figure 4 suggests that the parent GCMs dominatethe inter-model distances for both d0 and d1 for temperature, but perhaps not for precipitation where there is clear structure from RCM pairs. This is perhaps one of the more interesting results from the paper - and the authors should make more of it. Why is this the case, what are the mechanisms? What recommendations would the authors give for end-users of CORDEX given this finding?

The mechanisms for different results for DJF tas over BI and JJA pr over EA are commented on in the paper (Section 5) :

*"It is clearly seen that when large-scale phenomena are responsible for output, as in case of temperature changes over BI region, RCMs tend to be very close to driving GCM, and different GCMs are apart from each other (Figs. 1 and 7). On the contrary, when smaller scale processes are more in play, such as in case of JJA precipitation changes over EA, the results are more influenced by RCMs (Figs. 2 and 8). This does not automatically imply any real added value in the sense of more realistic simulation. Rather, it points to differences in implementation of the local processes in different RCMs. In our case, different parameterization schemes employed to simulate convection, microphysical processes in clouds and surface processes including soil moisture are possible candidates."*

Our results are not really representative for air temperature and precipitation over the whole European domain and for all seasons, but it illustrates that there are large differences between individual cases. Therefore, a recommendation for end-users is that an analysis of GCM-RCM interactions and a thorough choice of representative simulations (if it is not possible to use the whole multi-model ensemble) for impact studies is necessary. Our paper offers a tool for such analysis. We added a comment on this into the last sections:

*"The results of presented case study for two basic climatic variables over two European regions show that the structure of the multi-model ensemble and the GCM-RCM interactions can differ substantially in individual cases. Therefore, before the RCM outputs are used in any applied research (e.g. studies on impacts of projected future climate changes) an analysis of GCM-RCM interactions and a thorough choice of RCMs to be used is necessary. Present paper offers a convenient tool for this purpose."*

---

## Author Comment (AC3) · 23 Oct 2018

Please find enclosed the revision of our manuscript. We have revised it based on the comments of both anonymous referees, please find detailed description of the changes made to the manuscript in the replies to their comments. The revisions made are visible in the revised document, only the changes made to Fig. 4 and 5 are not highlighted. The Supplement now includes three parts, one is the original Supplement with the code for all the calculations, the second part are two figures illustrating the dependence of the d0 and d1 distances on the smoothing parameter and the third part includes figures related to the analysis of uncertainty connected to internal climate variability, as explained in the text of the paper.

[Figure]

Please also note the supplement to this comment:
https://www.geosci-model-dev-discuss.net/gmd-2018-157/gmd-2018-157-AC3-supplement.zip

―――――――――――――――――――――――

---

## Author Response (AR2)

We are grateful for the evaluation of our paper and the acceptance subject minor revision. We have revised the manuscript accordingly and accepted all the suggested revisions. Please find below our responses including the description of the changes made to the manuscript. The revisions made are visible in the revised document below.

The Supplement now includes two parts, one is the original Supplement with the code for all the calculations, the second part includes figures related to the analysis of uncertainty connected to internal climate variability, as explained in the text of the paper.

**Responses to comments of the Editor**

1) The figures showing the effect of different smoothing on d0 and d1 included in supplement 2 should be included in the main paper, rather than relegated to a supplement, at the very least because analysis of sensitivity to smoothing ought to be a standard step in any work on distance metrics.

The figures have been added to the paper as Figs. 4 and 5. The numbering of all subsequent figures was changed accordingly. There is a new sentence in the Methodology section "The analysis of sensitivity to amount of smoothing was carried out."

The numbering of Figs. in Supplement 2 (formerly Supplement 3) was also changed.

2) The discussion in the authors' response to reviewer 2's first comment about not using spatial information in the metric is interesting and valuable. This should be added to the paper, as it provides a good rationale for the focus on temporal information, and sheds some light on the difficulties of using spatial fields, especially in RCMs.

A new paragraph including the discussion was added to the last Section of the paper.

3) In addition to these two changes, please clarify the sentence on page 10, line 9: "Quite straightforward and clearly interpretable results suffered…". It's not clear whether you're saying the results were straightforward and interpretable (but suffered additional weaknesses), or whether the weaknesses were big enough that the results were are not straightforward nor easily interpretable.

The sentence was changed, we believe not it is clear: "Their results were Qquite straightforward and clearly interpretable, but 
[revised manuscript text omitted]